# Hierarchical Procedural Meta-Reasoning for Generalizable Multimodal Agents

Yao Fu [1]   Shengyi Qian [2]   Pierluca D'Oro [2]   Fanyi Xiao [2]   Honglak Lee [1 3]   Joseph Tighe [2]   Manchen Wang [2]

## Abstract

While multimodal agents can achieve strong performance through fine-tuning, their ability to generalize remains limited in complex real-world tasks such as mobile navigation, where diverse applications, frequent system changes, and customized workflows are common in practice. We argue that a fundamental bottleneck lies in whether an agent possesses sufficient task-specific procedural knowledge to accomplish a given goal. Such procedural knowledge may be provided by the general capabilities of large language models, or obtained from additional external resources such as web search when necessary. Based on this view, we propose **P**rocedure-**A**ware **M**ultimodal **A**gent with Meta-**R**easoning, a framework that explicitly represents task knowledge as natural-language procedures and trains a procedure-aware grounded agent to condition its actions on this knowledge. By learning to leverage procedural knowledge from different sources, our approach enables robust generalization across tasks, applications, interface versions, and multi-app workflows, achieving substantial improvements on challenging Android benchmarks.

## 1. Introduction

There has been growing interest in building multimodal agents due to their ability to map rich multimodal observations to low-level actions in interactive environments (Liu et al., 2024b; Zheng et al., 2024; Furuta et al., 2023). Although recent foundation models fine-tuned on task-specific data have substantially improved perception and action generation (Furuta et al., 2023; Durante et al., 2024), achieving robust generalization in such settings remains an open challenge. In particular, a major factor limiting generalization is the absence of sufficient task-level knowledge that

specifies how a goal should be accomplished under novel conditions (Agashe et al., 2024; Shlomov et al., 2024).

In this work, we study this challenge in the context of Android mobile navigation, which serves as a particularly suitable testbed for investigating generalization. Real-world mobile applications are highly diverse and exhibit rapidly evolving interfaces, user-specific configurations, and time-sensitive workflows. As a result, despite the availability of large-scale, high-quality Android interaction datasets (Rawles et al., 2023; Sun et al., 2025a; Li et al., 2020; Chai et al., 2025), agents fine-tuned for Android execution continue to struggle to generalize to realistic mobile usage scenarios (Sun et al., 2025a).

A key limitation of existing mobile agents is their reliance on implicitly encoded task knowledge. Agents trained primarily from interaction trajectories excel at grounded actions, but tend to entangle task understanding with application-specific action patterns observed during training. This coupling makes them brittle under distribution shifts, as the agent lacks an explicit representation of how a goal should be achieved beyond the training distribution. Therefore, we consider task procedures as an explicit form of task knowledge, consisting of temporally abstract and human-interpretable steps that describe how a goal can be accomplished. Conditioning low-level action generation on such procedures provides global task structure for grounded execution. Indeed, we find that procedures generated by large language models already constitute a strong source of knowledge at test time and substantially improve generalization.

However, procedural knowledge inferred by language models is not always sufficient. In realistic mobile settings, it can be outdated or mismatched, particularly for scenarios that postdate the training data of the LLM. Blindly following an incorrect procedure can therefore lead to systematic failure (Taioli et al., 2025; Wang et al., 2025) rather than graceful recovery. We may always invoke external information sources, such as web search or user clarification, whenever uncertainty arises. However, unconditional retrieval is fundamentally misaligned with interactive mobile agents for several reasons. First, external search interrupts closed-loop perception and action, introducing latency and breaking temporal coherence during execution. Second, retrieved information may be noisy, overly specific, or incompatible

[1]University of Michigan [2]Meta [3]LG AI Research. Correspondence to: Yao Fu <violetfy@umich.edu>.

*Proceedings of the 43rd International Conference on Machine Learning*, Seoul, South Korea. PMLR 306, 2026. Copyright 2026 by the author(s).

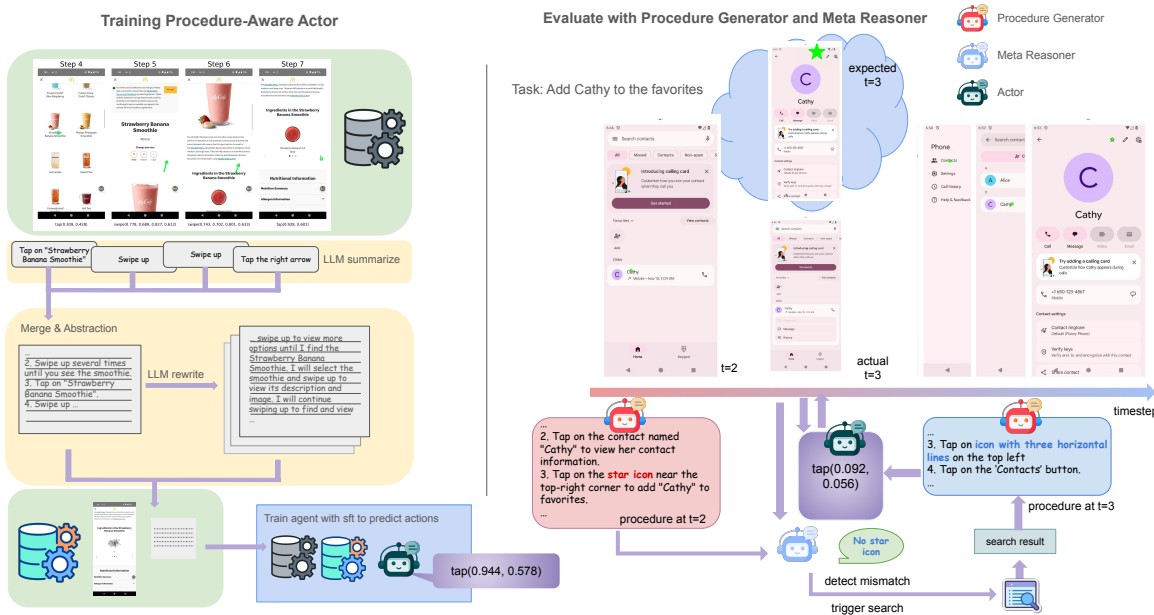

*Figure 1.* Overview of **P**rocedure-**A**ware **M**ultimodal **A**gent with Meta-**R**easoning (PAMAR). Left: Training. We train a procedure-aware grounded actor using supervised fine-tuning on a mixture of interaction trajectories augmented with induced natural-language procedures and their paraphrases, enabling the actor to reliably condition low-level actions on task-level procedural guidance. Right: Test time. The procedural generator produces and updates a task-level procedure online based on the current goal, observations, and interaction history. A meta-reasoning module evaluates whether the maintained procedure remains sufficient to explain ongoing execution; when a procedural deficiency is detected, external information retrieval is selectively triggered to help refine the procedure. Otherwise, execution proceeds using the internally updated procedure. [1]

with the current interface state, potentially overriding correct internal knowledge with brittle instructions. Moreover, always-search policies eliminate the need for the agent to reason about the limits of its own procedural understanding. By collapsing all uncertainty into retrieval, the agent never explicitly distinguishes between cases where its existing knowledge is sufficient and cases where it is not.

This motivates a central question: can an agent explicitly assess whether its current procedural knowledge is *sufficient to reliably complete a task*? To this end, we propose **P**rocedure-**A**ware **M**ultimodal **A**gent with Meta-**R**easoning (PAMAR), a procedure-aware multimodal agent that explicitly reasons about the sufficiency of its task-level knowledge during execution. PAMAR represents task understanding as natural-language procedures and decouples high-level procedural reasoning from low-level grounded action generation. Crucially, the agent is equipped with a meta-reasoning mechanism that continuously evaluates whether its current procedure remains adequate given the observed interaction context, and selectively triggers external information acquisition only when procedural deficiencies are detected. This design enables the agent to leverage internal procedural knowledge when it is sufficient, while robustly incorporating external information when it is not, leading to improved generalization under distribution shifts.

Our contributions are summarized as follows:

- We introduce *procedural sufficiency* as a key abstraction for multimodal agents, and propose a meta-reasoning mechanism that explicitly evaluates whether the agent's current procedural knowledge is adequate for task completion, enabling selective external information acquisition.

- We propose an LLM-guided training framework that induces temporally abstract natural-language procedures from demonstrations to guide low-level action generation, decoupling high-level procedural reasoning from grounded execution to improve generalization.

- Our proposed method demonstrates substantial generalization improvements on challenging Android benchmarks, including unseen applications, post-training interface changes, and multi-app workflows.

## 2. Related Work

**Multimodal Mobile Agents.** Recent advances in large foundation models and the availability of extensive Android task datasets (Zhang et al., 2023; Wang et al., 2024;

---

[1]The robot icon is made by Freepik from Flaticon.com, and the dataset icon is made by bsd from Flaticon.com. The windows search icon is provided by Icons8.

Rawles et al., 2023; Sun et al., 2025a) have driven significant progress in mobile agents that interact with real-world interfaces (Xu et al., 2024). While these agents excel on in-distribution tasks, their generalization to unseen applications remains limited (Gu et al., 2026), highlighting a persistent challenge in GUI-based agent research (Pan et al., 2024; Deng et al., 2023). Studies show that the trained agents often overfit to specific layouts and workflows, leading to poor performance on unseen or structurally different interfaces (Deng et al., 2023; Gu et al., 2026). Increasing model size or dataset diversity can further improve results on seen apps but has little effect on cross-application generalization (Sun et al., 2025a). Motivated by these limitations, we focus on improving generalization by introducing explicit procedural knowledge and training the agent to follow such procedures when its internal knowledge is insufficient.

**Exploration and External Search for Agents.** To mitigate the performance limitations arising from insufficient task knowledge in large foundation models, a growing body of work augments agents with exploration mechanisms (Sun et al., 2025b; Kulkarni et al., 2016) or external information sources (Zhang et al., 2023; Mei et al., 2025). When combined with large language models, external search improves task success on certain complex or long-horizon tasks. However, most existing methods invoke exploration or search unconditionally, either as a pre-execution phase (Agashe et al., 2024) or as a fixed component of the agent pipeline (Zhang et al., 2023). In contrast, we focus on enabling agents to selectively seek external information through explicit meta-reasoning, allowing them to balance internal knowledge with external guidance based on task familiarity.

**Hierarchical and Procedure-Guided LLM Agents.** Hierarchical decomposition is a widely adopted strategy for long-horizon decision making, where complex goals are broken down into intermediate subgoals or plans to improve reliability and reduce compounding errors (Chane-Sane et al., 2021; Nachum et al., 2018). Recent LLM-based agents extend this paradigm by generating high-level plans that guide stepwise execution in interactive environments (Xiong et al., 2025; Prasad et al., 2024; Sun et al., 2023; Song et al., 2022), including GUI and web settings (Erdogan et al., 2025; Liu et al., 2024a). While these approaches provide useful global structure, they primarily treat plans as execution scaffolds and focus on refinement or replanning in response to execution failures. In contrast, our work emphasizes *procedural sufficiency* as a distinct capability for agent control. Rather than using step-by-step knowledge solely for action decomposition, our agent maintains a complete task-level procedure and explicitly evaluates whether the remaining steps are plausible under the current UI context. This enables anticipatory identification of procedural deficiencies and selective triggering of external knowledge acquisition,

a capability that is largely absent from prior hierarchical or procedure-guided agent architectures.

# 3. Method

## 3.1. Background

We study multimodal agents operating in Android environments as a representative and challenging experimental setting for real-world interaction. A foundation-model-based agent is tasked with completing a natural-language goal $g$ by interacting with the Android user interface.

At each timestep $t$, the agent observes the current UI state as a screenshot $o_t$ and executes an action $a_t$ (e.g., tap, swipe, or text input), conditioned on the goal and the interaction history. Such an agent can be modeled as

$$a_t = \mathrm{LMM}(g, o_t, a_1, \ldots, a_{t-1}), \tag{1}$$

where LMM is typically fine-tuned on expert demonstrations using supervised fine-tuning (SFT) (Rawles et al., 2023; Sun et al., 2025a). While these agents achieve strong performance on tasks and apps seen during training, their generalization to novel applications and distribution shifts remains limited. In this work, we aim to improve generalization under this standard training and evaluation paradigm.

## 3.2. Procedure-Aware Multimodal Agent with Meta-Reasoning

We propose **P**rocedure-**A**ware **M**ultimodal **A**gent with Meta-**R**easoning (PAMAR), a multimodal agent framework that improves generalization by explicitly decoupling procedural knowledge reasoning from grounded action execution. As illustrated in Figure 1, PAMAR consists of three components: (i) a high-level procedural generator $M_p$, (ii) a meta-reasoning module $M_r$ that evaluates procedural sufficiency and governs external knowledge acquisition, and (iii) a low-level grounded actor $M_a$ that translates abstract procedures into concrete UI actions, each instantiated as a multimodal foundation model.

During execution, the procedural generator $M_p$ produces a task-level procedure, updating it online as the interaction unfolds. The meta-reasoning module $M_r$ evaluates procedural sufficiency and determines when external search is required, providing additional information to refine the procedure when triggered. The grounded executor $M_a$ conditions on the current procedure to execute UI actions sequentially.

## 3.3. Procedure-Aware Grounded Executor

Let $p$ denote a task-specific procedure: an ordered list of natural-language steps specifying how to accomplish the goal. This procedure provides an explicit, human-interpretable representation of the agent's task understand-

ing and supplies global structure for grounded execution. The low-level actor model $M_a$ interacts directly with the environment by producing concrete UI actions given $p$.

### 3.3.1. TRAINING WITH PROCEDURAL SUPERVISION.

We train the grounded low-level executor $M_a$ with supervised fine-tuning (SFT) on procedure-augmented data:

$$a_t = M_a(g, o_t, a_1, \ldots, a_{t-1}, p), \qquad (2)$$

so that it can reliably follow procedures when available. Conditioning on the full procedure throughout execution provides a global task structure that helps the executor ground individual actions in context.

### 3.3.2. PROCEDURE INDUCTION AND LANGUAGE AUGMENTATION

To construct procedural supervision, we synthesize compact, temporally abstract natural-language procedures from expert demonstrations. Details are shown in Appendix B.2.

**Transition-grounded step generation.** Given a demonstration trajectory $\{(o_{t-1}, a_t, o_t)\}_{t=1}^T$, we first produce a fine-grained description for each atomic UI action. For each timestep, we prompt an LLM with the pre-action screenshot $o_{t-1}$, the executed action $a_t$, and the post-action screenshot $o_t$, and extract a concise text description $d_t$ of the action intent grounded in the observed UI transition.

**Step merging and normalization.** The per-timestep descriptions $d_t$ are often redundant due to retries and micro-actions. We therefore canonicalize surface form (e.g., normalizing tense and removing boilerplate) and merge consecutive descriptions that reflect the same short-horizon intent. For example, we merge repeated swipes and collapse retry taps by identifying sequences with the same screenshots where the tap coordinates are within a small pixel radius.

**Condition-guided abstraction.** To better match how humans express procedures and to reduce sensitivity to action counts, we further rewrite action-centric instructions into intent- and condition-guided forms when applicable. For example, we replace fixed-count scroll directives (e.g., *swipe up twice*) with state-dependent stopping conditions (e.g., *swipe a few times until the desired element appears*).

**Procedure language augmentation.** Finally, to improve robustness to linguistic variation at test time, we generate paraphrases for each procedure by prompting an LLM to rewrite the step descriptions we got from previous steps in alternative natural-language forms while preserving semantics. During training, we sample from the original induced procedure and its paraphrases when conditioning the grounded

executor, exposing it to heterogeneous procedure styles (e.g., terse numbered instructions vs. narrative guidance).

### 3.4. High-Level Procedural Reasoner

The high-level procedural reasoner is responsible for producing and maintaining task-level procedural knowledge during execution. It consists of two components: a procedural generator and a meta-reasoning module.

### 3.4.1. PROCEDURAL GENERATOR

At test time, the procedural generator $M_p$ produces an initial natural-language procedure conditioned on the goal $g$ and the initial observation. It then dynamically updates the procedure online at each timestep based on the current observation and interaction history, allowing it to adapt to interface changes and correct potential mistakes. Prompts are provided in Appendix B.3. Crucially, the agent is prompted to maintain a complete procedure throughout execution, enabling task-level sufficiency judgments about whether the remaining steps are plausible under the current context.

### 3.4.2. PROCEDURE META-REASONING

At each timestep $t$, the meta-reasoning module $M_r$ assesses whether the procedure from the last step $p_{t-1}$ remains valid and sufficient to achieve the task goal $g$ given the interaction history $\{o_1, a_1, \ldots, o_{t-1}, a_{t-1}\}$, and outputs whether the agent can reliably proceed with it.

When $M_r$ determines that $p_{t-1}$ is not a good procedure and is not good enough to recover easily, it triggers an external knowledge acquisition tool call (e.g., web search). The retrieved information is then passed back to the procedural generator $M_p$ to update the procedure. This selective mechanism enables anticipatory knowledge augmentation while avoiding unnecessary tool use when the internally maintained procedure remains adequate.

We study two approaches for implementing this decision process: a prompt-based method that uses human-designed heuristics, and a training-based method.

**Prompt-Based Meta-Reasoning.** We implement $M_r$ using a structured reasoning prompt that guides stepwise analysis over the ongoing episode with human heuristics. The prompt instructs $M_r$ to assess progress with respect to $p_{t-1}$, verify whether the outcome of recent actions is consistent with expected UI changes, and check whether required UI elements for upcoming steps are present or reachable. Based on this assessment, $M_r$ outputs a decision to either continue with the current knowledge or trigger external acquisition. Prompts can be found in Appendix B.4.

**Training-Based Meta-Reasoner.** In addition to prompting, we train $M_r$ to predict when external acquisition is required. During training, we consider a case in which the external search provides perfect information for $M_p$ to generate the ground-truth procedure. Under this assumption, we use the induced expert procedure $p^\star$ from Section 3.3.2 as the reference signal. We then prompt $M_p$ to iteratively and dynamically generate a per-timestep procedure $\hat{p}_t$ under the same interaction context. This process yields an internal procedure that reflects the intrinsic task knowledge encoded in $M_p$ as well as the information inferred from observations.

We derive supervision signals by comparing how well $M_a$ explains expert actions under the two procedures. Specifically, let $\ell_t(p)$ denote the log-probability of generating the ground-truth action $a_t^\star$ under procedure $p$:

$$\ell_t(p) = \log P_{M_a}\left(a_t^\star \mid g, o_t, a_{<t}, p\right). \quad (3)$$

We then define the margin $\Delta_t = \ell_t(p^\star) - \ell_t(\hat{p}_t)$. A large positive margin indicates that the reference procedure explains the expert behavior substantially better than the internally generated procedure, suggesting a procedural deficiency that motivates triggering external acquisition. Concretely, we label a timestep as requiring search when the action induced under $\hat{p}_t$ is incorrect, the action induced under $p^\star$ is correct, and $\Delta_t$ exceeds a fixed margin threshold.

# 4. Experiments

## 4.1. Experimental Setup

**Datasets.** We evaluate PAMAR along three complementary dimensions of generalization.

First, we evaluate the agents on **DigiData-Bench-Auto** (Sun et al., 2025a), which assesses both cross-task generalization on seen applications and cross-application generalization on previously unseen apps.

Second, we introduce **DigiData-AppUpdate**, which evaluates robustness to application version changes, including deprecated functionalities and UI/workflow shifts relative to the training data.

Third, we introduce **DigiData-MultiApp**. Agents are trained on single-app objectives, while evaluation episodes require completing tasks across multiple apps, stressing long-horizon planning and app transitions.

**Baselines.** Our primary baseline is an 8B-parameter Perception–Language Model, **PLM-8B** (Cho et al., 2025), fine-tuned for Android control. We follow the supervised fine-tuning (SFT) recipe of DigiData (Sun et al., 2025a) and train on a mixture of four datasets: DigiData (Sun et al., 2025a), AitW (Rawles et al., 2023), AndroidControl (Li et al., 2020), and Cauldron (Laurençon et al., 2024).

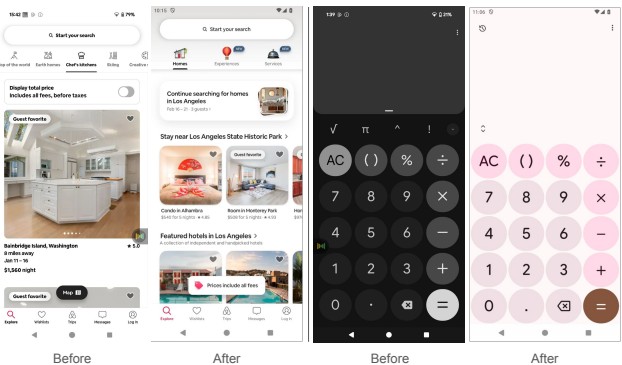

*Figure 2.* Examples of application version updates in DigiData-AppUpdate. Left: An Airbnb update removes category shortcuts below the search bar (e.g., *Top of the world*), breaking procedures that rely on these entry points. Right: A calculator redesign changes layout and visual structure while preserving core functionality, illustrating workflow/UI drift.

At test time, the policy predicts the next action given the task goal $g$, the current screenshot observation $o_t$, and the action history $a_{1:t-1}$: $a_t = \text{PLM}(g, o_t, a_{1:t-1})$. The action space (e.g., tap, type, swipe) is defined in Appendix A.1.

In addition, we evaluate GPT-4o as a direct action generator.

**Procedure-Augmented Agents.** We train PLM-8B as the low-level grounded actor $M_a$ using procedure-augmented trajectories, as described in Section 3.3. To isolate the effects of (i) conditioning execution on explicit procedures, (ii) enabling online procedure revision, and (iii) selectively acquiring external procedural knowledge, we evaluate three agent variants. All variants use GPT-4o as the procedural generator.

- **PAMAR-Static.** The agent generates a single procedure based on task goal and initial observation at the beginning of the episode using only internal knowledge, and follows this fixed procedure throughout execution.

- **PAMAR-Dynamic.** The agent may revise the procedure online using only internal knowledge, conditioning on the current observation and interaction history to correct UI references and adapt the workflow during execution.

- **PAMAR-MR-Dynamic.** In addition to online procedure revision, a meta-reasoner continuously assesses whether the current procedure is sufficient for task completion. External information acquisition (e.g., web search) is triggered only when a procedural deficiency is detected, after which the procedural generator produces an updated procedure that incorporates the retrieved information.

For the prompt-based meta-reasoner, we use GPT-4o with a structured reasoning prompt. For the training-based meta-reasoner, we fine-tune a Llama-4-Scout-17B-16E model

*Table 1.* Results on DigiData-Bench-Auto. The reported numbers are success rate (%) on cross-task, cross-app generalization tasks, and the whole evaluation set. Search(%) denotes the fraction of episodes that invoke external knowledge acquisition.

| Method | Agent model | Cross-Task | Cross-App | All | Search(%) |
|---|---|---|---|---|---|
| Direct (no procedure) | PLM-8B | 46.93 | 31.39 | 41.47 | N/A |
| Direct (no procedure) | GPT-4o | 11.80 | 3.93 | 9.00 | N/A |
| PAMAR-Static | PLM-8B | 49.83 | 39.88 | 46.30 | N/A |
| PAMAR-Dynamic | PLM-8B | 58.47 | 40.51 | 52.10 | N/A |
| PAMAR-MR-Dynamic (Prompt) | PLM-8B | 58.10 | 47.06 | 54.20 | 45.14 |
| PAMAR-MR-Dynamic (Train) | PLM-8B | 56.99 | 47.06 | 53.47 | 79.86 |
| Always-search | PLM-8B | 62.40 | 41.17 | 54.90 | 100.0 |

using the supervision described in Section 3.4.2. Additional implementation details are provided in Appendix B.4.

**App documentation gathering.** External acquisition is performed via app-specific knowledge search. Given the task goal $g$, we prompt an LLM to generate search queries targeting app-specific how-to instructions. When available, we query the official help center of the target app; otherwise, we fall back to FAQ-style support pages. Retrieved webpages are captured as screenshots to preserve multimodal information. The retrieved content is then processed by a knowledge accumulation module, which converts visual signals into textual descriptions and extracts task-relevant procedural snippets with an overlapping sliding-window scheme. In practice, the retrieved information is not assumed to be perfectly accurate and may be incomplete, partially relevant, outdated, or even misleading.

### 4.2. Cross-Task and Cross-Application Generalization

**Setup.** We evaluate cross-task and cross-application generalization using DigiData-Bench-Auto (Sun et al., 2025a), which tests whether agents can transfer procedural knowledge to task goals and application contexts that differ from those observed during training. Due to the nature of this benchmark, where tasks have clearly specified goals and relatively short horizons (at most 25 interaction steps), we restrict agents to searching at most once per episode. When search is triggered, the retrieved information remains available for all subsequent steps in the trajectory.

**Results.** Table 1 reports success rate and the fraction of episodes that invoke search. Conditioning the grounded actor on explicit task procedures improves generalization, with consistent gains in both cross-task and cross-application settings. Notably, even a static procedure generated once at the start helps, suggesting that explicit procedural structure provides global guidance beyond memorized action sequences; updating procedures online with internal knowledge (PAMAR-Dynamic) yields additional gains, especially

for cross-task generalization, indicating benefits from adapting task-level structure during execution.

External information further strengthens generalization: both always-search and MR agents outperform procedure-only variants overall, implying that retrieval supplies useful task- and application-specific knowledge in unfamiliar tasks and unseen applications, beyond what the procedural generator alone provides. At the same time, selective acquisition is substantially more efficient. Compared to unconditional retrieval, PAMAR-MR-Dynamic reduces search usage from 100% to 45.14% of tasks, while maintaining comparable success.

Finally, the training-based meta-reasoner exhibits more conservative search behavior than the prompt-based variant, triggering acquisition more frequently. This is expected because training assumes retrieved information is high-quality and consistently helps recover expert-induced procedures, biasing the policy toward searching under procedural ambiguity. In addition, training-time supervision compares action likelihoods to a single reference trajectory in a static dataset; since multiple valid action sequences may solve the same goal, reasonable deviations can be penalized, further encouraging search. Despite this tendency, the results show that procedural sufficiency signals can be learned from data.

### 4.3. Application Version Update Adaptation

**Setup.** Mobile applications are frequently updated, often introducing changes that invalidate previously correct task procedures. This is also true for applications in DigiData-Bench-Auto (Sun et al., 2025a). Since the original DigiData demonstrations were collected, many target apps on the emulators have undergone version updates that break or alter existing workflows. To isolate this source of distribution shift, we construct **DigiData-AppUpdate**. More specifically, we identify two prevalent update patterns: *functionality deprecation*, where a previously supported feature is removed, and *workflow or UI drift*, where the task remains supported but its interface or navigation structure changes. Examples

*Table 2.* Results on DigiData-AppUpdate. SR(%) denotes success rate. Search(%) is the fraction of episodes that query external update notes. Procedures are generated with GPT-4o.

| Method | SR(%) | Search(%) |
|---|---|---|
| No procedure | 29.50 | N/A |
| PAMAR-Static | 30.35 | N/A |
| PAMAR-Dynamic | 55.75 | N/A |
| PAMAR-MR-Dynamic (Prompt) | 59.03 | 39.30 |
| PAMAR-MR-Dynamic (Train) | 63.93 | 78.69 |

*Table 3.* DigiData-MultiApp Results. These tasks require navigation across two applications. SR(%) denotes the percentage of successfully completed episodes, while Search(%) denotes the percentage of episodes that require querying external information.

| Method | SR(%) | Search(%) |
|---|---|---|
| PLM-8B (No procedure) | 0.0 | N/A |
| PAMAR-Static | 0.0 | N/A |
| PAMAR-Dynamic | 10.0 | N/A |
| PAMAR-MR-Dynamic | 15.0 | 60.0 |

are shown in Figure 2. From the DigiData training pool, we select 61 training task goals on seen apps whose original completion trajectories are affected by such updates and evaluate them on updated app versions.

In this setting, external information corresponds to update-specific information, such as migration hints or release notes. Because reliable public documentation is unavailable for most tasks, we provide curated human-written update notes as the external retrieval results for this experiment.

**Results.** Table 2 reports success rates and search frequencies on DigiData-AppUpdate. The action-only baseline performs poorly, confirming that post-training application updates can invalidate previously learned execution patterns even on seen apps. Adding a static procedure provides little benefit, as the LLM-based procedure generator itself may also encode outdated procedural knowledge.

Allowing dynamic procedure revision substantially improves success, indicating that conditioning high-level guidance on the current UI state can partially compensate for interface drift. Incorporating external update notes yields the highest performance: PAMAR-MR-Dynamic recovers from cases where critical task knowledge is missing due to deprecated features or altered workflows, while avoiding unnecessary retrieval by triggering external acquisition only when procedural insufficiency is detected.

The training-based meta-reasoner retrieves external knowledge more frequently than the prompt-based variant, but also achieves higher success. This behavior is expected, as our training objective assumes that retrieved information reliably enables recovery of the expert procedure, biasing the learned policy toward conservative retrieval. In this setting, where external knowledge is human-curated, the apps are seen during training, and many interface structures and UI elements remain unchanged after the update, such behavior leads to improved task completion.

### 4.4. Compositional Multi-App Generalization

**Setup.** Real-world mobile objectives frequently require coordinating information and actions across multiple appli-

cations (e.g., retrieving information from system settings and acting on it in a browser). However, most Android agent training data consists of single-app trajectories, encouraging agents to overfit to within-app workflows and making app-to-app transitions particularly challenging.

To evaluate this setting, we construct **DigiData-MultiApp**, a focused benchmark of 20 compositional tasks that require completing a goal across two applications. Each episode requires extracting task-relevant information from a source app and subsequently using a target app to complete a dependent action. For example, *in Phone by Google, find which carrier is currently used for outgoing calls; then open Google Chrome and search for deals for that carrier.*

**Results.** Results are shown in Table 3. Single-step action policies fail completely in this setting, indicating that agents trained solely on single-app trajectories struggle to initiate app transitions or preserve intermediate information across applications. Adding a static procedure does not improve performance, as a fixed plan is insufficient for long-horizon tasks that require long-term memory. Allowing dynamic procedure revision enables some success, demonstrating that adapting high-level guidance during execution can partially support cross-app reasoning. Although absolute performance remains low due to the difficulty of the benchmark, PAMAR-MR-Dynamic achieves the highest success rate, suggesting that explicit procedural guidance and selective retrieval can be beneficial even in compositional multi-app settings. Notably, the meta-reasoner continues to invoke retrieval selectively rather than defaulting to retrieval in every episode, indicating that retrieval decisions remain controlled as task complexity increases.

### 4.5. Ablation

#### 4.5.1. TRAINING–TEST PROCEDURE ALIGNMENT

We observe that grounded procedure following should be learned rather than imposed purely by prompting at test time. In particular, if the executor is trained only on action-only demonstrations, providing a procedure at test time may not help and can even be harmful, as the model is not calibrated to interpret procedural language. Moreover, even when

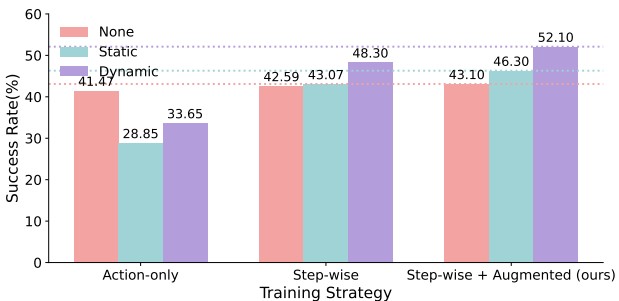

*Figure 3.* **Ablation Study: Training–Test Procedure Alignment** Success rates (%) on DigiData-Bench-Auto for three evaluation settings: no procedure (None), static procedure (Static) and dynamic procedure (Dynamic). All procedures are generated with GPT-4o.

training includes procedures, robustness may depend on the diversity of procedural language seen during training.

To test these hypotheses, we vary the form of procedural supervision during training and evaluate with and without procedures at test time. Concretely, we compare: (i) action-only training, (ii) training with step-wise procedures generated from demonstrations with simple step merging and normalization, and (iii) training with step-wise procedures plus our full procedure abstraction and augmentation. At evaluation time, we measure success on novel applications under two conditions: without procedures (standard actor setting) and with procedures (procedure-conditioned execution).

Figure 3 highlights two findings. First, when the executor is trained without procedural supervision, adding procedures at test time does not reliably help and can hurt performance, consistent with a train–test mismatch in inputs. Second, step-wise procedural supervision improves procedure-conditioned execution, but a large gap to the fully augmented procedure set remains, suggesting sensitivity to procedure phrasing/format. Further augmentation yields the biggest gains, indicating that exposure to diverse procedural variants is important for robust test-time following.

### 4.5.2. META-REASONER TRIGGER TIMING

Our framework relies on the meta-reasoner to decide *when* external procedural knowledge should be acquired. To test whether the *timing* of these triggers matters, we perform a controlled test-time intervention that delays retrieval.

**Intervention.** For the training-based meta-reasoner, when the model predicts that external acquisition should be triggered, we suppress the *first* such trigger in an episode with 50% probability, forcing the agent to continue execution using its internally maintained procedure. To avoid trivially removing retrieval for the remainder of the episode, we add a fallback: if no retrieval is executed for the next 5 decision steps, we force a retrieval call at the next step. All other

*Table 4.* **Ablation: Meta-Reasoner Trigger Timing.** Success rate on DigiData-Bench-Auto (All split). Delaying the first predicted retrieval trigger noticeably degrades performance.

| Method | SR(%) |
|---|---|
| PAMAR (Train) | 53.47 |
| PAMAR (Train, Delayed Trigger) | 48.61 |

components, including the procedural generator, actor, and the retrieval mechanism itself, are kept identical.

**Results.** The delayed-trigger intervention reduces success rate on DigiData-Bench-Auto from 53.47% to 48.61% (Table 4). The performance drop indicates that *when* retrieval is performed matters: delaying acquisition can cause the agent to enter incorrect UI states or take hard-to-reverse actions, after which later retrieval becomes less effective. Notably, this result is complementary to the strong performance of the *always-search* variant reported in Table 1, which shows that external knowledge can be highly beneficial. However, always-search indiscriminately performs retrieval even when the internally maintained procedure is already sufficient, which can lead to unnecessary tool usage and latency in interactive settings. Taken together, these results suggest that effective deployment of retrieval requires not only access to external knowledge, but also appropriate trigger timing.

## 5. Conclusion

In this work, we presented PAMAR, a procedure-aware multimodal agent that improves generalization by explicitly reasoning about the sufficiency of task-level procedural knowledge during execution. By decoupling high-level task knowledge from low-level grounded action generation and selectively acquiring external information only when internal procedures are insufficient, our approach avoids the brittleness of unconditional retrieval. Experiments on challenging Android navigation tasks demonstrate substantial generalization gains across unseen tasks and applications, interface changes, and multi-app workflows, while reducing unnecessary external tool usage. These results highlight procedural sufficiency as a useful abstraction for building robust interactive agents.

## Impact Statement

This work studies multimodal agents that interact with real-world mobile interfaces by reasoning about task-level procedures and selectively acquiring external information. Such agents have the potential to improve accessibility, productivity, and usability by assisting users with complex or unfamiliar mobile workflows. However, at the same time, several societal risks warrant consideration. Like other learning-based systems, our approach depends on offline datasets

and language model priors, which may encode biases. If deployed without appropriate safeguards, agents could make suboptimal decisions or provide misleading guidance. In addition, autonomous agents capable of interacting with mobile applications may be misused for malicious purposes, such as automating spam, unauthorized data access, or deceptive behaviors.

Mitigating these risks requires responsible development and deployment practices, including incorporating human oversight and user-in-the-loop mechanisms, improving transparency around agent decision-making and external information use, and enforcing security and access controls to prevent misuse. We emphasize that our work is intended only as a research contribution toward understanding procedural reasoning and selective information acquisition. We believe that ongoing engagement with the broader research community and relevant stakeholders is essential to ensure that advances in foundation model–based agents remain aligned with societal well-being and ethical considerations.

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

---

**Algorithm 1** PAMAR Test-Time Execution with Meta-Reasoned Knowledge Acquisition

---

**Require:** Goal $g$, environment $\mathcal{E}$, procedure generator $M_p$, meta-reasoner $M_r$, grounded actor $M_a$, knowledge search tool Search$(\cdot)$, knowledge processor Accumulate$(\cdot)$

1: Observe initial screenshot $o_0 \leftarrow \mathcal{E}.\text{reset}(g)$
2: Initialize procedure $p \leftarrow M_p(g, o_0)$
3: Initialize external knowledge buffer $K \leftarrow \emptyset$
4: **for** $t = 0, 1, 2, \ldots, T$ **do**
5:     **Sufficiency check:** $d_t \leftarrow M_r(g, o_{\leq t}, a_{<t}, p)$ $\{d_t \in \{\text{continue}, \text{search}\}\}$
6:     **if** $d_t = \text{search}$ **then**
7:         Generate app-specific query $q_t \leftarrow \text{QueryGen}(g, p, o_t)$
8:         Retrieve pages/screenshots $S_t \leftarrow \text{Search}(q_t)$
9:         Extract knowledge $k_t \leftarrow \text{Accumulate}(S_t)$
10:        Update buffer $K \leftarrow K \cup \{k_t\}$
11:     **end if**
12:     **Procedure update:** $p \leftarrow M_p(g, o_t, a_{<t}, p, K)$
13:     **Grounded action:** $a_t \leftarrow M_a(g, o_t, a_{<t}, p)$
14:     Execute $a_t$, observe next screenshot $o_{t+1} \leftarrow \mathcal{E}.\text{step}(a_t)$
15:     **if success or termination condition then**
16:         **break**
17:     **end if**
18: **end for**

---

# A. Evaluation Settings

## A.1. Evaluation Framework

**Dynamic auto-evaluation.** We evaluate all agents using a dynamic automatic evaluation framework based on realistic Android emulators set up in DigiData-Bench-Auto. Each episode runs inside an emulator with the target application installed and initiated with a pre-defined emulator image, and the environment state is reset at the beginning of each trial.

**Success measurement with LLM-as-a-judge.** As many tasks have open-ended natural language goals and multi-step UI interactions, by default, we determine task success using an LLM judge protocol with GPT-4o, which was established by the DigiData benchmark (Sun et al., 2025a).

**Action Space** Agents interact with the environment through the following actions:

- `tap(x, y)`: Tap at screen location $(x, y)$.

- `swipe(x1, y1, x2, y2)`: Swipe from $(x_1, y_1)$ to $(x_2, y_2)$.

- `type(text)`: Type a string into the currently focused input field.

- `navigate(option)`: System navigation actions with `option` $\in \{\text{back}, \text{home}, \text{enter}\}$.

- `status(option)`: Terminate an episode with `option` $\in \{\text{complete}, \text{impossible}\}$.

## A.2. Datasets

We evaluate PAMAR along three complementary generalization axes.

### A.2.1. TASK AND APPLICATION GENERALIZATION.

We use DigiData-Bench-Auto (Sun et al., 2025a), which evaluates generalization across tasks and applications by grouping evaluation apps into *seen*, *familiar*, and *novel* categories with respect to the training set. We follow the standard protocol of (Sun et al., 2025a) for environment initialization, task definitions, and success evaluation.

A.2.2. APPLICATION VERSION UPDATE ADAPTATION.

To isolate distribution shift caused by post-training application updates, we construct DigiData-AppUpdate. Starting from the DigiData training task pool, we identify 61 goals on *seen* applications whose original successful trajectories no longer execute reliably due to version changes. We then re-evaluate the same goals on the updated app versions and group update-induced failures into two common patterns: (i) functionality deprecation, where a previously available feature is removed or renamed, making the original workflow infeasible; and (ii) workflow/UI drift, where the task remains supported but the interface or navigation changes (e.g., layout changes, deeper menus, or relocated controls). Figure 2 shows representative examples of both patterns.

A.2.3. SINGLE-TO-MULTI APP GENERALIZATION (DIGIDATA-MULTIAPP).

To evaluate whether agents trained only on single-app trajectories can generalize to multi-app workflows, we construct DigiData-MultiApp, a benchmark of 20 tasks that require composing actions across two applications. Each task requires (i) obtaining a piece of information or completing a subgoal in a source app, and (ii) switching to a target app to perform a follow-up action conditioned on the result from the source app. Agents are trained on single-app objectives only and never observe multi-app demonstrations during training.

# B. Procedure-Aware Multimodal Agent with Meta-Reasoning

## B.1. Model Training

We fine-tune PLM-8B using supervised fine-tuning (SFT). Training is conducted in two stages. In the first stage, we train the model for 60K iterations on base data without procedures. In the second stage, we continue training for an additional 15K iterations on a mixture of data with and without procedures. Unless otherwise specified, we use a learning rate of $2 \times 10^{-5}$ and a batch size of 2 samples per GPU. We fine-tune the full model, including the vision encoder and adaptation layers. We use an image tile size of $448 \times 448$ and limit each screenshot to at most four tiles. Training is performed on 128 NVIDIA A100 GPUs.

## B.2. Procedure Synthesis

This section details the procedure induction and language augmentation pipeline used to construct procedural supervision from expert demonstrations. In DigiData (Sun et al., 2025a), each demonstration is a length-$T$ trajectory of observation–action pairs $\{(o_t, a_t)\}_{t=1}^{T}$, where $o_t$ is a screenshot and $a_t$ is an atomic UI action (e.g., tap, swipe, type). Our goal is to convert such low-level trajectories into compact, temporally abstract natural-language procedures that can be used to condition the grounded executor during training.

B.2.1. TRANSITION-GROUNDED STEP EXTRACTION

We first convert each atomic action into a short natural-language step grounded in the observed UI transition. For each timestep $t$, we prompt an LLM with: (i) the pre-action screenshot $o_{t-1}$, (ii) the executed action $a_t$ (including tap/swipe coordinates), and (iii) the post-action screenshot $o_t$. The model outputs a structured summary of the transition; we extract from this output a concise action-intent description and denote it as a per-timestep description $d_t$. We use Prompt STEP SUMMARIZATION PROMPT for this purpose, which is based on the Chain-of-Thought generation prompt from DigiData (Sun et al., 2025a).

---

**STEP SUMMARIZATION PROMPT**

You are an expert in summarizing the progress of an Android navigation agent, whose role is to help a human user navigate the Android phone to complete a goal. The agent takes step-by-step phone actions like clicking, swiping, typing, navigating home, navigating back, and ending.

The agent's action at this step is {action}. If the action involves tapping or long press, the tap location is shown on screen A with a hollow green circle. If the action involves swiping, the start and end points of the swipe are shown on Screen A with a green arrow. {xml_string}

First summarize screen A, screen B. Focus on:

- What key elements (e.g., icons, texts, buttons, navigation bars, images, forms and input field, pop-ups, switches and toggles etc) are there in each screenshot and where are they (e.g., top-left, bottom-right etc).

- Do NOT mention the terms "screen A" or "screen B" in your response. Instead, refer to them as the screenshot.

Next describe the agent's action. Make it short, in one sentence, and use future tense. Focus on:

- What does the agent do to get from A to B. If the action involves tapping, infer which key element is tapped based on the action, hollow green circle location and the context of the screenshot. If the action involves swiping, infer the swipe direction based on the action, green arrow and the context of the screenshot.

- Do NOT mention the terms "hollow green circle" or "green arrow" in your response.

Then summarize the changes from A to B. Use future tense. Focus on:

- What are the changes from A to B related to the goal. Pay attention to subtle changes including altered or new key elements.

- Do NOT mention the terms "screen A" or "screen B" in your response. Refer to screen A as the current screenshot and screen B as the next screenshot.

Finally provide a reasoning on why does the agent take this action. Use future tense. Focus on:

- What are the reasons for the agent to take this action? Why does this action help the agent to achieve the goal?

Be concise and comprehensive. Only mention key elements which are related to the goal and ignore the irrelevant ones.

Use the first-person pronoun "I" as if you are the agent.

Your answer should be in the following format, ensuring each section is no more than 100 words:

  screen A: <summary of screen A>

  screen B: <summary of screen B>

  action description: <action description>

  changes: <changes from A to B>

  reason: <why took this action>

---

### B.2.2. STEP MERGING AND NORMALIZATION

The per-timestep descriptions $\{d_t\}$ are fine-grained and often contain redundant retries or micro-actions (e.g., repeated scrolls or taps). We therefore merge consecutive descriptions into a shorter sequence of higher-level steps $\{s_k\}_{k=1}^K$ using Android-specific heuristics:

- **Action-form normalization.** Before comparing or merging step descriptions, we canonicalize their surface form by

removing tense/inflectional variants, common prepositions, and personal pronouns that do not affect intent. This yields a normalized representation that better reflects the underlying UI intent.

- **Tap equivalence under stable observations.** For tap actions, we treat two consecutive steps as equivalent if (i) the pre-action observations are identical and (ii) the tap coordinates are within a small pixel radius. Intuitively, repeated taps on the same screen at approximately the same location typically correspond to the same intended UI interaction.

- **Action consolidation.** Using the normalized action forms and the tap-equivalence rule above, we merge consecutive steps with the same inferred intent into a single abstract instruction. For repeated actions, we express repetition explicitly (e.g., *Tap X a few times* or *Scroll down several times*).

### B.2.3. Condition-Guided Abstraction

To better reflect how humans express procedures, we further rewrite action-centric steps into intent- and condition-guided subgoals whenever applicable. For example, we replace fixed-count scroll instructions (e.g., *Swipe up twice*) with state-dependent forms (e.g., *Scroll up until you see the desired element*). This transformation encodes both the intent and an explicit stopping condition, improving robustness to UI variability and reducing sensitivity to exact action counts. We also standardize completion markers (e.g., issue *status(complete) to complete the task*) to produce more consistent procedure structure.

### B.2.4. Procedure Language Augmentation

To improve robustness to linguistic variation at test time, we generate paraphrases for each induced procedure. Concretely, we prompt an LLM to rewrite the merged procedure into alternative natural-language forms while preserving its semantics. In our implementation, we provide the model with the sequence of step descriptions and ask it to produce a more narrative, high-level version of the same procedure.

---

**NARRATIVE PROCEDURE REWRITING PROMPT**

You are an expert in summarizing the progress of an Android navigation agent, whose role is to help a human user navigate the Android phone to complete a goal. The agent takes step-by-step phone actions like clicking, swiping, typing, navigating home, navigating back, and ending.

You are given a trajectory that aims to achieve the goal of "{goal}". There are {total_steps} steps in this trajectory and you are currently at step {step_index}. For each step, you are provided with a description of the screenshot before agent's action and the agent's action.

First, understand the progress by summarizing what has already happened based on the step-by-step descriptions below of what the agent has done before step {step_index}:

{history_string}

Next, summarize what will happen as a procedure based on the step-by-step descriptions below of what the agent will do in and after step {step_index}:

{future_string}

Please follow these guidelines when providing your response:

- Do NOT mention specific step numbers.

- Follow the step-by-step descriptions closely when summarizing the progress and the procedure.

- Use the first-person pronoun "I" as if you are the agent.

Your answer should be in the following format, with each section limited to 100 words or fewer:

    Summary: <summary of all previous steps>

    procedure: <procedure of all future steps (including current one)>

---

## B.3. Procedure Generation

The procedure generator in this work is based on GPT-4o unless otherwise specified. Prompts for procedure generation and procedure update are shown below.

---

**PROCEDURE GENERATOR**

You are an expert in creating step-by-step procedures for Android navigation tasks. Your role is to provide complete, detailed instructions that would help someone who has never done this task before to accomplish their goal from start to finish. You must first reason through the task by analyzing what needs to be done, then provide a comprehensive step-by-step procedure.

Please follow these guidelines when providing your response:

- CRITICAL: Each step must correspond to exactly ONE atomic action. The agent can only perform one of these actions per step: <action space description >

- Never let the agent log in or update apps.

- UI ELEMENT DESCRIPTION: The bottom-left corner is reserved for the 'Navigate Back' button. When referring to other elements near the bottom-left, describe their appearance only (icon/text), or explicitly say 'near the corner but NOT the navigate back button'.

- RESOLVE CONDITIONS:

Please provide a step-by-step procedure to accomplish the given goal from start to finish, as if creating instructions for someone who has never done this task before. The procedure should start from the beginning of the workflow when you open the app.

Example 1:

Goal: Use Settings app, Disconnect from currently connected Wi-Fi networks

Step-by-Step Procedure:

1. Swipe up a few times until you see Settings icon.

2. Tap on the Settings icon.

3. Tap on "Internet".

4. Tap on the Wi-Fi toggle switch to turn it off.

5. Tap on "Done".

6. Issue "status(complete)" to stop the task.

Example 2:

...

Example 3:

...

Now, provide the step-by-step procedure for the following goal:

Goal: "goal"

Your answer should be in the following format:

Step-by-Step Procedure:

1. [First step]

2. [Second step]

3. [Third step]

...

N. [Final step]

---

---

PROCEDURE UPDATE PROMPT

Review the current screenshot and action history to refine the existing step-by-step procedure if needed.
IMPORTANT: Even when updating the plan based on the current screen, you must provide a COMPLETE procedure as general knowledge - starting from the very beginning when the user first opens the app. Do NOT provide a partial plan that only covers remaining steps. The procedure should serve as a comprehensive guide that works for anyone attempting this task from scratch.
Instructions:
Compare the current screen with the previous plan. Make minor adjustments only if:
- A UI element name differs from expected (e.g., `"My Account"` instead of `"Account"`)
- The screen state reveals a different path is needed
If the plan is still accurate, return it unchanged.
First, reason through what adjustments (if any) are needed based on the current screen state.
- Although you should generate the whole procedure, pay attention to the current observation/history to correct any existing errors.
- Ignore any update/log in steps.
- UI ELEMENT DESCRIPTION: The bottom-left corner is reserved for the 'Navigate Back' button. When referring to other elements near the bottom-left, describe their appearance only (icon/text), or explicitly say 'near the corner but NOT the navigate back button'.
- The last step must be Issue `"status(...)"` to stop the task
Here are some examples of step-by-step procedures we want from you, notice that it should always be starting from the very beginning:
Example 1:
Goal: Use Settings app, Disconnect from currently connected Wi-Fi networks
Step-by-Step Procedure:
1. Swipe up a few times until you see Settings icon.
2. Tap on the Settings icon.
3. Tap on `"Internet"`.
4. Tap on the Wi-Fi toggle switch to turn it off.
5. Tap on `"Done"`.
6. Issue `"status(complete)"` to stop the task.
Example 2: ...
Example 3: ... Now, for the following goal, first provide your reasoning about any needed adjustments, then provide the complete step-by-step procedure (starting from app launch).
Goal: `"{goal}"`
Previous Plan:
`{prev_plan}`
Action History:
`{action_history}`
Your answer MUST follow this exact format, DO NOT put anything after the last step of the procedure:
Reasoning:
[Your analysis of the current screen and any adjustments needed]
Step-by-Step Procedure:
1. [First step - should typically involve opening/launching the app]
2. [Second step]
3. [Third step]
...
N. [Final step]

## B.4. Meta Reasoner

Meta reasoning prompt is shown below.

META REASONER

You are an expert Android task execution assistant. Your role is to analyze the current task progress by carefully tracking where the agent is in its plan, and decide whether external knowledge is needed to proceed effectively.
You will be given:
1. The goal/task to accomplish
2. The plan you generated last step (step-by-step procedure)
3. The action history (what has been done so far)
4. The most recent two screenshots
Your job is to determine if the agent is on the right track or if external knowledge is needed.
Analyze the current task execution and decide if external knowledge is needed.
Goal: {goal}
Current Plan:
{current_plan}
Action History:
{action_history}
Instructions - Follow this reasoning process step by step:
**Step 1: Locate Current Position in Plan**
- Compare the current observation and action history against the current plan and identify which step number the agent has just completed or is currently executing.
**Step 2: Verify Previous Action Outcome**
- Compare the current screen with the previous screen (you have access to the most recent images).
- Did the last action executed match the plan?
- Did that action produce the expected change in the UI?
**Step 3: Assess Next Step & Plan Validity**
- Is the UI element or action required for the *next* step visible/available on the current screen?
- **If YES:** The plan is working. ⇒ **CONTINUE**
- **If NO:** Analyze WHY:
- **Case A (Minor Mismatch / Easy Fix):** The specific element is missing, BUT there is an obvious alternative that achieves the same sub-goal (e.g., button is named slightly differently, or there's a clear path forward). The agent's general knowledge is correct, just needs a minor plan adjustment. ⇒ **CONTINUE**
- **Case B (Wrong Plan / Knowledge Gap):** The previous action didn't work as expected (e.g., tapped a button but nothing happened, or went to a wrong screen), OR the next step is missing and there is NO obvious path forward. This implies the agent's understanding of the procedure is incorrect. ⇒ **SEARCH**
Your response MUST follow this exact format:
Reasoning:
Step 1 - Position: [Where is the agent in the plan?]
Step 2 - Prev Action Check: [Did the last action work as expected? Explain based on screen changes.]
Step 3 - Analysis: [Is the next step achievable? If not, is it a minor mismatch (CONTINUE) or a wrong plan (SEARCH)?]
Decision: [CONTINUE or SEARCH]

