# OpenReview forum: "Hierarchical Procedural Meta-Reasoning for Generalizable Multimodal Agents"
_ICML.cc/2026/Conference — ICML 2026 regular_

### Official Review · Reviewer_Mv6Y · 2026-03-08

**Soundness:** 3
**Presentation:** 3
**Significance:** 3
**Originality:** 3
**Overall Recommendation:** 4
**Confidence:** 3

**Summary:**

This paper proposes PAMAR, a multimodal Android agent framework that explicitly represents task knowledge as natural-language procedures and separates high-level procedural reasoning from low-level grounded UI action generation. The framework has three main parts: a procedural generator that produces and updates task procedures online, a grounded actor trained to execute actions conditioned on those procedures, and a meta-reasoner that decides when the current procedure is insufficient and external knowledge retrieval should be triggered. The paper also introduces a procedure induction pipeline that turns expert demonstrations into abstract procedural supervision for training. Empirically, the method is evaluated on three settings: cross-task/cross-app generalization on Digidata-Bench-Auto, robustness to app version changes in Digidata-AppUpdate, and compositional multi-app tasks in Digidata-MultiApp. The reported results suggest that explicit procedural conditioning and selective retrieval improve generalization over direct action prediction baselines.

**Compliance With Llm Reviewing Policy:**

Affirmed.

**Key Questions For Authors:**

1. How sensitive are the gains to the choice of procedural generator and meta-reasoner LLM? If the improvements rely heavily on GPT-4o specifically, that would affect my assessment of reproducibility and generality. A comparison with a weaker or open model would strengthen the claim that the framework itself, rather than a particular model, is responsible for the gains.


2. Can the authors better quantify the cost-benefit tradeoff of selective retrieval versus always-search? Since always-search is slightly stronger on the main benchmark, explicit measurements of latency, token/tool cost, and perhaps robustness to noisy retrieval would clarify when selective meta-reasoning is truly preferable in practice.

3. How robust is the training-based meta-reasoner to alternative valid procedures? Because supervision is derived from comparison against a single induced reference procedure, I would like to know whether the learned trigger policy over-searches in tasks with multiple correct workflows. A deeper analysis here could improve my confidence in the “procedural sufficiency” formulation.

4. Can the authors provide more detail on Digidata-AppUpdate and Digidata-MultiApp construction? In particular, how were tasks selected, how diverse are the update patterns, and what safeguards were used to avoid benchmark bias toward the proposed method? Stronger benchmark documentation would increase confidence in the empirical claims.

5. Why does static procedural guidance sometimes help only marginally or even hurt?A more systematic failure analysis separating outdated procedures, hallucinated steps, and actor-following errors would clarify where the main bottleneck lies and whether meta-reasoning solves the right failure mode.

**Limitations:**

yes

**Strengths And Weaknesses:**

Strengths:

- Overall, the work studies a critical problem. The authors strive to analyze a pertinent question: whether a multimodal mobile agent can explicitly determine when its current procedural knowledge is sufficient for execution and when it should seek external information. This is a meaningful problem setting for Android agents, where UI drift, unseen apps, and multi-app workflows make pure action imitation brittle.

- Among the main strengths, the paper presents a clear and intuitive decomposition of the agent into a procedural generator, a grounded actor, and a meta-reasoner. This decomposition is easy to understand and aligns well with the paper’s motivation. The idea of representing task knowledge as explicit natural-language procedures is sensible, and the procedure induction pipeline from demonstrations is a practical way to provide high-level supervision to the executor. The empirical setup is also reasonably aligned with the paper’s claims: the evaluation covers unseen tasks/apps, app version updates, and multi-app workflows, which together provide a relevant test of generalization. The reported improvements on Digidata-Bench-Auto and Digidata-AppUpdate are meaningful enough to suggest that procedure-conditioned execution is genuinely helpful beyond standard direct action prediction.

- Another strength is that the paper does more than propose a static planning scaffold. The dynamic-procedure and meta-reasoning components attempt to model when internal knowledge is insufficient, which is a useful step beyond simply always planning or always retrieving. I also found the ablations on train–test procedure alignment and trigger timing valuable, because they support two important intuitions: first, that procedure following must be learned rather than injected only at inference time; and second, that the timing of retrieval matters, not just its availability.


Weaknesses:
- On the other hand, the paper has several weaknesses that limit how strong the overall case currently is. The biggest one is that the core notion of “procedural sufficiency” remains only partially operationalized. In practice, the training-based meta-reasoner relies on action log-probability comparisons against a single induced reference procedure. This is a workable heuristic, but it does not fully establish that the model is detecting insufficiency in a robust conceptual sense, especially when multiple valid procedures may exist for the same task. As a result, part of the claimed conceptual novelty feels stronger in framing than in formalization.

- A second weakness is that it is not fully clear how much of the gain should be attributed to the proposed framework itself versus the use of strong external components at inference time. The procedural generator is GPT-4o, retrieval is supported by external app documentation, and the best results often involve these additional capabilities. This does not invalidate the method, but it makes it harder to isolate the contribution of the framework design from the contribution of stronger test-time assistance. In particular, the fact that always-search slightly outperforms the selective variants on the main benchmark weakens the practical argument for meta-reasoning unless cost, latency, and robustness to noisy retrieval are analyzed more explicitly.

- The evidence for the hardest setting is also still limited. The multi-app benchmark is interesting, but with only 20 tasks and absolute success rates still very low, it is difficult to draw strong conclusions about compositional generalization. The benchmark is useful as an initial stress test, but not yet strong enough to fully substantiate broad claims about multi-app reasoning.

---

> ### Author Rebuttal · Authors · 2026-03-31
>
> Thank you for your insightful feedback. We now answer each feedback below and hope that our response addresses your concerns.
>
> > … A comparison with a weaker or open model would strengthen the claim that the framework itself...
>
> We have conducted some additional experiments with an open-sourced model, Llama 4 17B, on Digidata-Bench-Auto. The results are presented below:
>
> | Method| SR  |
> |-|-|
> | PAMAR-static|47.76%|
> | PAMAR-dynamic|52.46%|
>
> Based on the results, a weaker open-source model also brings substantial improvements, suggesting that the gains are not specific to GPT-4o but instead stem from the proposed training framework. Due to the limited time and compute during rebuttal, we have not completed all experiments. We will continue running evaluations and include them in a later version of the paper.
>
> > … better quantify the cost-benefit tradeoff of selective retrieval versus always-search? …
>
> Thank you for the insightful suggestion. Below is a comparison of search cost per task. As shown below, the retrieval pipeline is non-trivial, involving LLM-based query generation and iterative summarization of gathered screenshots. As a result, always-search (AS) incurs substantially higher latency and token usage, while selective retrieval effectively reduces both.
>
> ||Search Latency (s)| Search Tokens| Meta-Reasoning Tokens|
> |-|-|-|-|
> |PAMAR |310.8|73k|8k|
> |AS|683.6|162k|0|
>
> > How robust is the training-based meta-reasoner to alternative valid procedures? …
>
> We thank the reviewer for the insightful question. We agree that multiple valid procedures exist for a task, but in our framework their impact is partially mitigated. First, we compare the log-probability of generating the ground-truth action, rather than strict matching to a single reference procedure. The procedural generator dynamically updates procedures to adapt to states, and we also explicitly train the agent to be more robust to procedures through extensive data augmentation. In particular, the procedure-augmented data includes LLM-rewritten variants that introduce slight imperfections such as hallucination and ambiguity. This encourages the agent not to blindly follow the procedure, but also rely on its own grounded judgment when necessary. Second, the procedure generator does not consistently produce a single deterministic procedure. Instead, sometimes it generates conditional procedures, multiple alternative strategies, or higher-level abstract instructions. This further reduces the reliance on a single canonical workflow during both training and inference. Incorporating retrieval-in-the-loop training, as well as leveraging multiple valid reference trajectories, is a promising direction for future work and could further improve the calibration of retrieval decisions.
>
> > … more detail on Digidata-AppUpdate and Digidata-MultiApp construction? …
>
> For Digidata-AppUpdate, tasks are manually selected from the training data to ensure that they are indeed affected by app updates.
> For Digidata-MultiApp, we construct tasks in two ways. First, half of the tasks are formed by composing combinations of existing training tasks, where we design coherent workflows that connect multiple apps. Second, the remaining tasks introduce new, unseen tasks, while requiring interaction across two apps. This setup ensures both compositional generalization and task novelty across applications.
>
> > Why does static procedural guidance sometimes help only marginally or even hurt? … whether meta-reasoning solves the right failure mode…
>
> Procedural steps generated at the initial timestep cannot fully anticipate all subsequent states and transitions during execution. The procedures can gradually become insufficiently grounded to the environment, leading to failures like looping for 38.3% of failed trajectories. On the other hand, Search-grounded plans tend to be more grounded, giving the actor model enough signal to try a valid approach rather than repeating the same failed actions. There are 16.4% of the tasks where search triggered early, substantially changed the plan, and the new plan was correct. This suggests that meta-reasoning helps address failures caused by insufficient or mismatched procedural knowledge. While this brings improvements, due to search quality and action following errors, the performance of PAMAR-MR-Dynamic is still not perfect. We provide a failure analysis of PAMAR-MR-Dynamic in the table below. This analysis shows that meta-reasoning significantly reduces loop-type failures, indicating improved procedural grounding.
>
> | Stuck Loop ↓ | Wrong Exec ↓ | Search: Wrong/No Knowledge ↓ | Search: Too Late ↓ | Search: Correct Knowledge, Actor Failed ↓ |
>  |-|--|-|-|-|
> | 1.60% | 11.50%| 32.80%| 8.20% | 45.90% |

---

> > ### Author Rebuttal · Reviewer_Mv6Y · 2026-04-04
> >
> > Thanks for the response, I will keep my positive score.

---

### Official Review · Reviewer_StX2 · 2026-03-12

**Soundness:** 3
**Presentation:** 3
**Significance:** 3
**Originality:** 3
**Overall Recommendation:** 4
**Confidence:** 3

**Summary:**

This paper studies the generalization challenge of multimodal agents in complex environments such as Android mobile navigation. The authors argue that a key bottleneck is the lack of explicit procedural knowledge describing how a goal should be accomplished. To address this, they propose a Procedure-Aware Multimodal Agent with Meta-Reasoning, which represents task knowledge as natural-language procedures and conditions low-level actions on these procedures. The framework includes hierarchical reasoning to select or refine procedures from different sources (e.g., language models or external retrieval). Experiments on Android navigation benchmarks show improved generalization across tasks, applications, and multi-app workflows.

**Compliance With Llm Reviewing Policy:**

Affirmed.

**Final Justification:**

My concerns are basically addressed, and I will maintain my initial justification.

**Key Questions For Authors:**

-How robust is the agent when the generated procedure is partially incorrect or outdated? Do you observe cascading failures in such cases?

-Can the authors clarify how the meta-reasoning module decides when to rely on LLM-generated procedures versus retrieving external information?

-Have you tested the framework in other multimodal agent settings beyond Android navigation (e.g., web agents or robotics tasks)? If so, how transferable is the approach?

**Limitations:**

yes

**Strengths And Weaknesses:**

Strengths

+The paper identifies an important limitation of current multimodal agents—implicit task knowledge embedded in trajectories—and proposes an explicit procedural representation to improve generalization.

+The hierarchical design that separates procedural reasoning from low-level action generation is intuitive and aligns well with how complex tasks are structured.

+The experimental evaluation on Android mobile navigation is relevant and demonstrates consistent improvements over existing mobile agents, suggesting the approach is practically useful.

Weaknesses

-The core idea of representing task execution as natural-language procedures and conditioning actions on them is conceptually related to prior work on planning or instruction-guided agents, and the novelty mainly lies in the specific framework design.

-It is somewhat unclear how robust the meta-reasoning component is when procedures are partially incorrect or outdated; more analysis of failure cases would strengthen the claims.

-The evaluation focuses primarily on Android environments, so it remains unclear how well the approach generalizes to other multimodal agent domains.

---

> ### Author Rebuttal · Authors · 2026-03-31
>
> Thank you for your valuable feedback. We now answer each feedback below and hope that our response addresses your concerns.
>
> > How robust is the agent when the generated procedure is partially incorrect or outdated? …
>
> At test time, both retrieval results and generated procedures can be noisy, partially relevant, or even misleading due to factors such as retrieval errors, outdated or irrelevant online resources, and LLM generation issues (e.g., hallucinations or incomplete summaries). Here is an example of incorrect search results (summarized by LLM):
>
> Task: Use Wish app, Change country/region to Colombia
>
> Knowledge: You must adjust the currency settings on the non-mobile Wish website. Follow these steps:\n\n1. From a non-mobile device, open your web browser … The displayed language in the Wish app will match your phone's preferred language…
>
> Despite this, our method still demonstrates performance improvements. We attribute this to the fact that our agent is explicitly trained to be robust to imperfect procedures to some extent. In particular, our training pipeline includes procedure-level data augmentation, where LLM-rewritten procedures introduce variations such as ambiguity, abstraction, or mild inconsistencies. The grounded actor is therefore trained to produce correct actions under diverse and imperfect procedural inputs, which improves robustness at test time.
>
>
> > … it remains unclear how well the approach generalizes to other multimodal agent domains.
>
> We agree that evaluating generalization beyond Android is important. While we do not include cross-domain experiments in this work, we design our framework to be domain-agnostic. We chose the mobile setting as a representative and challenging testbed, which already captures key difficulties such as dynamic interfaces, diverse tasks, and long-horizon task execution.
>
> Importantly, our action design is not necessarily simple. The grounded actor operates over generic coordinate-based actions, which are more low-level, more general, and harder to ground than UI-element-based actions, giving them strong potential to generalize directly to diverse domains. In addition, our retrieval component is inherently web-based and therefore directly applicable to many other domains such as web navigation without modification.
>
> At the same time, conducting comprehensive evaluations across additional domains would require substantial effort, including collecting domain-specific search results and training and evaluating new models. Empirically validating this broader generalization is an important direction for future work.
>
> > Can the authors clarify how the meta-reasoning module decides when to rely on LLM-generated procedures versus retrieving external information?
>
> The detailed design of meta-reasoner is described in Sec. 3.4.2 (Procedure Meta-Reasoning).  Concretely, we study two approaches for implementing this decision process: (1) a prompt-based method that relies on human-designed heuristics, and (2) a training-based method that learns the decision from data. The prompt used in the first approach is provided in Appendix B.4. For the second approach, the full training details can be found in Sec. 3.4.2 (paragraph “Training-Based Meta-Reasoner”). In both cases, the meta-reasoner evaluates whether the current procedure is sufficient for the task, and triggers external retrieval when it is deemed insufficient.

---

### Official Review · Reviewer_6SZU · 2026-03-12

**Soundness:** 3
**Presentation:** 3
**Significance:** 3
**Originality:** 3
**Overall Recommendation:** 5
**Confidence:** 3

**Summary:**

This paper studies how multimodal agents can generalize beyond the tasks, applications, and workflows seen during training. The authors argue that a key limitation of current agents is the lack of explicit task-level procedural knowledge. To address this, they propose PAMAR (Procedure-Aware Multimodal Agent with Meta-Reasoning), which separates high-level procedural reasoning from low-level action generation and represents task knowledge as natural-language procedures induced from demonstrations. A meta-reasoning module evaluates whether the current procedure is sufficient for completing the task and selectively triggers external knowledge retrieval when necessary. The grounded actor conditions its action policy on these procedures and is trained using procedure-augmented supervision. The method is evaluated on Android interaction benchmarks covering cross-task and cross-application generalization, application updates, and multi-app workflows. Results show that explicit procedures and dynamic updates improve success rates over direct-action baselines. At the same time, selective retrieval achieves benefits similar to those of always-search strategies with fewer external queries.

**Compliance With Llm Reviewing Policy:**

Affirmed.

**Final Justification:**

The rebuttal addressed my main concerns.

**Key Questions For Authors:**

1. **Robustness to imperfect retrieval.**
   The meta-reasoner training assumes that retrieved information always leads to correct expert procedures. In real settings, search results can be noisy or outdated. Have the authors investigated how PAMAR behaves when external information is incorrect or conflicting? Would the meta-reasoner learn to distrust retrieval in such cases, and could this be incorporated into the training objective?

2. **Threshold sensitivity for search triggers.**
   The margin threshold used to trigger external search is fixed. How sensitive are the results to this threshold, and could a learned or adaptive threshold improve performance?

3. **Scalability of multi-step retrieval.**
   The current evaluation restricts agents to at most one external acquisition per episode. How would the framework handle tasks requiring multiple sequential clarifications, and what challenges arise from repeated retrieval in terms of latency, memory, and accumulation of potentially conflicting procedures?

4. **Generalization beyond Android.**
   The experiments focus on Android UI navigation. Does the proposed framework generalize to other domains such as web navigation or desktop GUIs? Are there specific properties of Android apps (e.g., relatively small action space) that make the approach more effective?

**Limitations:**

Yes

**Strengths And Weaknesses:**

### Soundness

**Strengths**
- The paper proposes a clear architecture comprising a procedural generator, a meta-reasoner, and a grounded actor.
- The procedure induction pipeline from demonstration trajectories is well-motivated and produces interpretable task guidance.
- Experiments cover multiple forms of generalization, including unseen apps, interface updates, and multi-app tasks.
- Ablations examine the impact of procedural supervision and retrieval timing.

**Weaknesses**
- The training of the meta-reasoner assumes that retrieved information always produces a correct procedure, which may not hold in practice.
- The search trigger threshold is fixed, and its sensitivity is not analyzed.
- Agents are limited to at most one retrieval per episode, which restricts evaluation of more complex tasks requiring iterative search.

---

### Presentation

**Strengths**
- The paper is well structured, with a clear motivation and method description.
- The procedure induction pipeline and training setup are explained clearly.

---

### Significance

**Strengths**
- Improving generalization for multimodal agents interacting with real software environments is an important research problem, and explicit procedural representations offer a promising approach to bridging high-level reasoning and grounded execution.
- The newly introduced benchmarks for app updates and multi-app workflows highlight important real-world challenges.

**Weaknesses**
- The evaluation focuses on Android environments, and it remains unclear how well the approach generalizes to other platforms or interaction domains.
- External retrieval is simulated using curated documentation, which may not fully reflect the real-world noise in web retrieval.

---

### Originality

**Strengths**
- The paper introduces the concept of procedural sufficiency and a meta-reasoner that evaluates whether an agent’s procedural knowledge is adequate during execution.
- The integration of procedure induction, meta-reasoning, and selective retrieval forms a coherent framework for improving generalization.

**Weaknesses**
- The novelty primarily lies in the integration and the meta-reasoning mechanism rather than in entirely new algorithmic components, as several individual components, such as LLM-generated plans and retrieval-augmented agents, have appeared in prior work.

---

> ### Author Rebuttal · Authors · 2026-03-31
>
> Thank you for your constructive feedback. We answer each feedback below and hope that our response addresses your concerns.
>
> > … how PAMAR behaves when external information is incorrect or conflicting? …
>
> We would like to clarify that the assumption that retrieval can recover the correct expert procedure is only made during training. At test time, both retrieval and procedures can be noisy or even misleading due to factors such as retrieval errors, outdated online resources, and LLM generation issues (e.g., hallucinations). Here is an example of incorrect LLM-summarized search results:
>
> Task: Use Wish app, Change country/region to Colombia
>
> Knowledge: You must adjust the currency settings on the non-mobile Wish website. Follow these steps:\n\n1. From a non-mobile device, open your web browser …
>
> Despite this, our method still improves performance. We attribute this to the fact that our agent is trained to be robust to imperfect procedures with procedure-level data augmentation, where LLM-rewritten procedures introduce variations such as ambiguity, abstraction, or mild inconsistencies. The grounded actor is therefore trained to produce correct actions under diverse and potentially imperfect procedural inputs, which improves test-time robustness.
>
> A more realistic setup would incorporate retrieved results during training and generate updated procedures accordingly. However, we fully respect the terms of service and data usage policies (e.g., robots.txt and licensing constraints) of external websites, and under our organizational policy, many web resources can be used for inference but not for training. Therefore, we could not conduct experiments under such a setting. Nevertheless, our current training approach can be viewed as a form of teacher forcing over high-quality procedures, while the augmentation pipeline exposes the agent to imperfect variants, partially addressing this gap.
>
> > … How sensitive are the results to this threshold…
>
> |Metric|0|2|3|
> |-|-|-|-|
> |search%|80|20|10|
> |success%|90|70|70|
>
> The results reported in the paper correspond to threshold = 0. We additionally evaluate models with threshold = 2 and 3. Due to limited time and compute resources during the rebuttal period, we are only able to evaluate these settings on a subset of 10 tasks. We observe that increasing the threshold significantly reduces the search rate, while also leading to a moderate drop in overall success rate, which is consistent with the intuition that external search provides useful information. The threshold values are chosen based on the empirical scale of sequence-level log-probability differences observed on the dataset. Learning or adapting the threshold dynamically based on context is a promising direction for future work.
>
> > …How would the framework handle… multiple sequential clarifications, and what challenges arise …?
>
> We would like to clarify that our retrieval pipeline involves LLM-generated queries, followed by web crawling and iterative LLM-based summarization. As such, a single retrieval is already relatively expensive and introduces the following overhead:
>
> | Metric| Value |
> |-|-|
> | Latency | 683.6s   |
> | Tokens |162k|
>
> In our benchmarks, most tasks are single-app and rely on a constrained retrieval source (e.g., app help centers). Additional attempts often provide limited new information. Moreover, LLM-generated queries can cover multiple sub-tasks, allowing one retrieval to address several aspects. We agree that more complex settings require multiple sequential retrievals. In our multi-app experiments, we allow retrieval separately for different apps, which generally does not lead to conflicting procedures. In this case, PAMAR still does not over-trigger search, refraining from retrieval in 8/20 tasks. This indicates that the meta-reasoner maintains selective triggering.
>
> > … Does the proposed framework generalize to other domains such as web navigation or desktop GUIs? …
>
> Our framework is generalizable to other GUI domains by design. We chose the mobile setting as a representative and challenging testbed, which captures key difficulties such as dynamic interfaces, diverse applications, and long-horizon task execution.
>
> Importantly, our approach is not Android-specific: The grounded actor generates generic coordinate-based actions, and our retrieval component is inherently web-based, enabling direct extension to domains such as web navigation.
>
> We also note that Android tasks are not necessarily simpler, as coordinate-based actions are lower-level, more general, and harder to ground than UI-element-based actions. We therefore expect the framework to generalize to other domains. That said, evaluating this would require substantial effort (e.g., new benchmarks, retrieval sources, and model training), and we leave this as important future work.

---

> > ### Author Rebuttal · Reviewer_6SZU · 2026-04-04
> >
> > Thanks for the author's response, and it resolved my concerns. I'll keep my current score.

---

### Official Review · Reviewer_DKe9 · 2026-03-13

**Soundness:** 3
**Presentation:** 3
**Significance:** 3
**Originality:** 3
**Overall Recommendation:** 4
**Confidence:** 4

**Summary:**

This paper studies generalization for Android multimodal agents and argues that a key bottleneck is whether the agent has sufficient task-level procedural knowledge for a goal. It proposes PAMAR, which represents task knowledge as natural-language procedures, trains a procedure-aware low-level actor on demonstration-induced procedures and paraphrases, and adds a high-level procedural generator plus a meta-reasoner that selectively triggers external retrieval when the current procedure appears insufficient. The empirical study covers three settings: cross-task/cross-app generalization on Digidata-Bench-Auto, robustness to post-training app updates on Digidata-AppUpdate, and a small multi-app benchmark. The reported results show clear gains over the no-procedure baseline on the first two settings, while the multi-app setting remains difficult.

**Compliance With Llm Reviewing Policy:**

Affirmed.

**Ethical Review Concerns:**

My concerns have been partially resolved. I will determine my final score after the further discussion with the AC and other reviewers.

**Final Justification:**

The authors have not fully resolved all of my concerns, but the rebuttal is constructive. I think the efficiency–robustness argument is more convincing than initially presented. I am willing to raise my score to 4, and I hope the final version more carefully situates the main claims in light of the remaining limitations on the main benchmark and the MultiApp setting. Good luck.

**Key Questions For Authors:**

1.On Digidata-Bench-Auto, always-search achieves the best overall success rate (55.28%), while meta-reasoned variants mainly reduce search frequency. Should the main claim be framed as improving efficiency rather than improving raw task success?

2.The trained meta-reasoner assumes that external retrieval can recover the correct expert procedure, and its supervision is based on comparison against a single induced reference procedure. How robust is this approach when retrieval is noisy, partially relevant, or misleading?

3.Digidata-AppUpdate uses curated human-written update notes because reliable public documentation is unavailable. Can the authors clarify how much of the reported gain depends on this curated setup, and whether similar improvements hold with automatically retrieved real web pages only?

4.The multi-app benchmark is promising, but the best result is still only 3/20. Can the authors provide a richer failure breakdown here (memory failure, bad app transition, wrong retrieval trigger, wrong procedure update, etc.) and clarify whether they see this benchmark as evidence of capability or mainly as evidence of current limitations?

**Limitations:**

yes

**Strengths And Weaknesses:**

**Strengths**

The motivation is clear: Android agents often overfit to action patterns and lack explicit task knowledge under distribution shift. The proposed decomposition into a procedural generator, a meta-reasoner, and a grounded actor is conceptually coherent, and the procedure-induction pipeline is reasonably well thought out, including step abstraction and language augmentation. The experiments are not limited to one benchmark split: the paper evaluates standard cross-task/cross-app transfer, app-update robustness, and a compositional multi-app setting, and it includes two helpful ablations on training–test procedure alignment and trigger timing.

**Weaknesses**

First, on the main benchmark, the strongest success rate is still achieved by the always-search variant (55.28%), not by either meta-reasoned variant (54.58% for prompt-based MR and 53.84% for trained MR). This means the paper most directly demonstrates an efficiency tradeoff—similar success with less search—rather than a clear raw-performance win from procedural sufficiency reasoning itself. Second, the training-based meta-reasoner relies on an idealized supervision scheme that assumes external search can recover the correct expert procedure and compares against a single induced reference trajectory; the paper itself notes that this biases the learned trigger toward conservative over-search. Third, the app-update benchmark uses curated human-written update notes as retrieval results, which is useful for diagnosis but weaker as evidence that the full retrieval loop would be robust in realistic noisy-document settings. Finally, the multi-app evidence is still quite limited: the best method solves only 3/20 tasks.

---

> ### Author Rebuttal · Authors · 2026-03-31
>
> Thank you for your helpful feedback. We answer each feedback below and hope that our response addresses your concerns.
>
> > … always-search achieves the best ... Should the main claim be … improving efficiency…?
>
> We agree that efficiency is an important benefit of PAMAR. Below is a comparison of search cost per task. The retrieval pipeline is non-trivial, gathering information with LLM-generated queries and iterative summarization. Always-search (AS) incurs substantial cost and latency while PAMAR is designed to balance the performance–efficiency tradeoff.
> |Method|Search Latency (s)| Search Tokens| Meta-Reasoning Tokens|
> |-|-|-|-|
> |PAMAR |310.8|73k|8k|
> |AS|683.6|162k|0|
>
> Moreover, as search can be noisy, AS is not consistently optimal. In the Cross-App split of Digidata-Bench-Auto, AS performs worse than PAMAR (41.17% vs. 47.06%), suggesting that unconditional retrieval can introduce noise that degrades performance. Taken together, we believe PAMAR has a strong advantage in representing knowledge as explicit procedures and selectively seeking external help when necessary. We will further emphasize this in a revised version of the paper.
>
> > The trained meta-reasoner assumes that external retrieval can recover ... How robust … when retrieval is noisy …?
>
> We would like to clarify that the assumption that retrieval can recover the correct expert procedure is only made at training. At test time, both retrieval results and generated procedures can be noisy or misleading due to retrieval errors and LLM hallucinations. An incorrect search result example:
>
> Task: Use Wish app, Change country/region to Colombia
>
> Knowledge: … Follow these steps:\n\n1. From a non-mobile device, open your web browser… Note: Currency settings may only be changed from the non-mobile site at this time…
>
> Despite this, our method still improves performance, as our agent is trained to be robust to imperfect procedures. In particular, our training pipeline includes procedure-level data augmentation, where LLM-rewritten procedures introduce ambiguity, abstraction, and mild inconsistencies. The grounded actor is trained to produce correct actions under diverse and imperfect inputs, improving test-time robustness.
>
> A more realistic setup would incorporate retrieved results during training. However, due to compliance with website terms of service (e.g., robots.txt) and organizational policies restricting the use of some web data for training, this is not feasible in our setting. Our current approach instead approximates this via teacher-forcing on high-quality procedures, with augmentation exposing the agent to imperfect variants.
>
> > Digidata-AppUpdate uses curated human-written update notes…
>
> For AppUpdate, our goal is to simulate a realistic access to app-specific update information(e.g. official update notes, release logs, or ask for expert help). We acknowledge that using curated human-written notes simplifies the retrieval and does not fully capture the real-world noise. We will clarify this limitation in the final version.
>
> Evaluating with real web retrieval is challenging because the considered updates are recent and often undocumented (e.g., UI restructuring or minor workflow changes). As a result, relying solely on web retrieval would conflate two causes of failure: the agent's inability to reason about UI changes, and the absence of relevant online information.
>
> However, we view the curated setup as a necessary diagnostic setting. By ensuring access to correct update information, it establishes a controlled environment that isolates the agent's reasoning capabilities—proving that when accurate knowledge about a UI drift is provided, our meta-reasoner can successfully update its internal procedure and recover from the failure. We believe demonstrating this capability in a controlled setting is a fundamental first step before addressing the additional challenges of noisy web retrieval.
>
> > The multi-app benchmark is promising… failure breakdown…
>
> We view the MultiApp benchmark as highlighting both current limitations and benefits of PAMAR. The low success rate mainly reflects limitations of the actor, which is trained only on single-app tasks thus struggles with cross-app transitions and long-horizon execution, leading to early termination. Nevertheless, explicit procedures enable non-trivial gains over the direct-action baseline, improving success from 0/20 to 3/20. Importantly, in this unfamiliar setting, PAMAR does not over-trigger search for different apps and is restrained from searching in 8/20 tasks, indicating that the meta-reasoner maintains selective triggering.
>
> Failures mainly fall into 3 types: cross-app transition errors, where the actor terminates prematurely; imperfect retrieval, with missing or misleading information; and imperfect procedure generation, with error accumulation over long horizons. We will include this analysis and clarify the benchmark’s role in evaluating long-horizon and multi-app generalization.

---

> > ### Author Rebuttal · Reviewer_DKe9 · 2026-04-04
> >
> > The efficiency argument is well-supported with concrete cost data, and the Cross-App result where PAMAR beats always-search is a fair point. However, always-search still achieves the best overall success rate on the main benchmark, and the multi-app results (3/20) remain too weak to support the generalization claims.

---

> > > ### Author Response · Authors · 2026-04-08
> > >
> > > Thank you for the follow-up comment. We appreciate the acknowledgment of the efficiency gains and the Cross-App results.
> > >
> > > Regarding the comparison with always-search (AS), we agree that AS achieves the highest absolute success rate on the main benchmark (AS 55.28% vs. ours 54.58%). However, we would like to clarify that our primary goal is not to always outperform AS in raw success, but to address a more practical setting where retrieval is costly and noisy. Specifically:
> > > - Efficiency: PAMAR reduces search usage from 100% to 45.46% of tasks, resulting in ~2× lower latency and token cost, while maintaining comparable success. In realistic deployments, this difference is substantial.
> > > - Robustness to unnecessary retrieval: As shown in the Cross-App split, AS underperforms PAMAR (41.17% vs. 47.06%), suggesting that unconditional retrieval can introduce noise and hurt performance.
> > > - Contribution of our training framework: Importantly, the gains of AS are not solely due to retrieval. Our procedure-aware training pipeline significantly improves the agent’s ability to follow procedures. As shown in Figure 3, removing this training leads to a substantial performance drop.
> > >
> > > Regarding the multi-app benchmark, we agree that this setting remains highly challenging, with current success rates still relatively low. We note that PAMAR already demonstrates strong performance on the main benchmarks (Digidata-Bench-Auto and Digidata-AppUpdate), and this more challenging multi-app setting is intended to further evaluate whether such gains extend to longer-horizon, compositional tasks, as well as whether the meta-reasoner continues to exhibit selective retrieval behavior. Despite the long-horizon difficulty, PAMAR still outperforms the direct-action baseline (0/20 to 3/20 successes), showing that explicit procedural guidance remains beneficial. Importantly, even as task complexity increases, the meta-reasoner does not degenerate into always triggering retrieval, maintaining selective and controlled retrieval behavior. We will more clearly explain the intended role of this benchmark and provide a more detailed analysis of the observed behaviors in the paper.
> > >
> > > ---
> > >
> > > We hope these clarifications help better contextualize our results and contributions, especially in terms of balancing performance and efficiency.

---

### Decision · Program_Chairs · 2026-04-30

**Decision:**

Accept (regular)

**Comment:**

This paper introduces PAMAR, a framework that improves the generalization of multimodal agents in Android benchmark by separating high-level procedural reasoning from low-level action execution. Reviewers appreciated the intuitive architectural decomposition and the comprehensive experimental setup that evaluated cross-task transfer, robustness to application updates, and multi-app workflows. While reviewers raised concerns regarding the meta-reasoner's reliance on idealized external retrieval and noted that selective search primarily benefits efficiency rather than raw success rates, the core methodology remains sound. The explicit modeling of procedural knowledge is a well-motivated approach that demonstrably enhances agent performance over standard direct-action baselines under various distribution shifts. Given the solid technical foundation and the introduction of highly relevant evaluation settings, the paper makes a meaningful contribution to the field and the AC recommends to accept the paper to ICML 2026.